# Lead Isotopic Constraints on the Provenance of Antarctic Dust and Atmospheric Circulation Patterns Prior to the Mid-Brunhes Event (~430 kyr ago)

**DOI:** 10.3390/molecules27134208

**Published:** 2022-06-30

**Authors:** Changhee Han, Laurie J. Burn, Paul Vallelonga, Soon Do Hur, Claude F. Boutron, Yeongcheol Han, Sanghee Lee, Ahhyung Lee, Sungmin Hong

**Affiliations:** 1Department of Ocean Sciences, Inha University, 100 Inha-ro, Michuhol-gu, Incheon 22212, Korea; hch@inha.ac.kr (C.H.); sanghee@kopri.re.kr (S.L.); leeahyung@inha.edu (A.L.); 2Division of Glacial Environment Research, Korea Polar Research Institute, 26 Songdomirae-ro, Yeonsu-gu, Incheon 21990, Korea; sdhur@kopri.re.kr (S.D.H.); yhan@kopri.re.kr (Y.H.); 3Department of Imaging and Applied Physics, Curtin University of Technology, GPO Box U1987, Perth, WA 6845, Australia; lburn81@outlook.com (L.J.B.); paul.vallelonga@uwa.edu.au (P.V.); 4Oceans Graduate School and UWA Oceans Institute, The University of Western Australia, Crawley, WA 6009, Australia; 5Institut des Géosciences de l’Environnement, Université Grenoble Alpes/CNRS, 54 rue Molière, 38400 Saint Martin d’Hères, France; claudeboutron@orange.fr; 6Research Unit of Frontier Exploration, Korea Polar Research Institute, 26 Songdomirae-ro, Yeonsu-gu, Incheon 21990, Korea

**Keywords:** lead isotopes, EPICA Dome C ice core, Mid-Brunhes Event, dust and volcanic sources, isotope mixing model, southern westerly winds

## Abstract

A lead (Pb) isotopic record, covering the two oldest glacial–interglacial cycles (~572 to 801 kyr ago) characterized by lukewarm interglacials in the European Project for Ice Coring in Antarctica Dome C ice core, provides evidence for dust provenance in central East Antarctic ice prior to the Mid-Brunhes Event (MBE), ~430 kyr ago. Combined with published post-MBE data, distinct isotopic compositions, coupled with isotope mixing model results, suggest Patagonia/Tierra del Fuego (TdF) as the most important sources of dust during both pre-MBE and post-MBE cold and intermediate glacial periods. During interglacials, central-western Argentina emerges as a major contributor, resulting from reduced dust supply from Patagonia/TdF after the MBE, contrasting to the persistent dominance of dust from Patagonia/TdF before the MBE. The data also show a small fraction of volcanic Pb transferred from extra-Antarctic volcanoes during post-MBE interglacials, as opposed to abundant transfer prior to the MBE. These differences are most likely attributed to the enhanced wet removal efficiency with the hydrological cycle intensified over the Southern Ocean, associated with a poleward shift of the southern westerly winds (SWW) during warmer post-MBE interglacials, and vice versa during cooler pre-MBE ones. Our results highlight sensitive responses of the SWW and the associated atmospheric conditions to stepwise Antarctic warming.

## 1. Introduction

The EPICA (European Project for Ice Coring in Antarctica) ice core drilled at Dome C (hereafter EDC) on the central East Antarctic Plateau (EAP) (75°06′ S, 123°21′ E, altitude 3233 m above sea level) has provided unique archives of past climate changes over the last successive eight glacial-interglacial cycles back to Marine Isotope Stage (MIS) 20.2, ~800 kyr before present (B.P.) [1,2]. The youngest four climate cycles are characterized by a larger amplitude of climate variability with warmer interglacials after the Mid-Brunhes Event (MBE), ~430 kyr ago, compared to the earlier smaller climate changes with relatively cooler interglacials [2,3]. Together with the Antarctic temperature record, dust records from deep Antarctic ice cores are of particular interest as indicators of the sensitivity of atmospheric and surface conditions in lower latitude dust source regions to glacial–interglacial climate change, affecting the dust cycle at high latitudes [4,5]. However, the dust flux data alone cannot be used to ascertain to what extent the southern westerly winds (SWW) shifted in response to climate cycles, which is considered one of the underlying mechanisms regulating glacial–interglacial variability of atmospheric CO_2_ [6,7]. A fundamental understanding of these paleo-atmospheric dynamics (that is, the latitudinal shift of the SWW) can be gained from study of climate-related dust provenance changes using isotopic and geochemical source tracers [8,9,10,11,12,13,14,15].

Strontium (Sr) and neodymium (Nd) isotopic compositions have been used for tracing the provenance of dust trapped in East Antarctic ice [8,9,10,11,12,13,14,15]. Compared to Sr-Nd isotope provenance studies, however, determination of the provenance of Antarctic dust using lead (Pb) isotopes has been limited to very few studies [16,17,18], although Pb isotopes can be used as fingerprints to constrain dust sources and their evolution through space and time [19]. In contrast to earlier Sr-Nd isotopic tracer studies, for which large sample volumes of >0.3 kg were required for high precision analysis in Antarctic glacial ice [8,9,10,11], the main advantage for a Pb isotope approach is that Pb isotopic compositions can be reliably determined in Antarctic ice using a sample weight of ~10 g, allowing for high-resolution records of Antarctic dust provenance changes in response to climatic conditions [16,17]. This highlights great potential for using Pb isotopes as an effective tracer of dust provenance and associated climate systems. Despite that, very limited Pb isotopic data have been reported for Antarctic ice dating back two glacial–interglacial cycles [16,17], with no availability of Pb isotope records prior to the MBE, which makes it difficult to examine the glacial–interglacial changes in the Southern Hemisphere atmospheric circulation under the influence of different climatic conditions before and after the MBE. This is mainly because a reliable Pb isotope measurement in Antarctic deep ice cores remains an analytical challenge due to extremely low Pb concentrations at or below the picogram per gram (10^−12^ g g^−1^) level and contaminants being brought to the outside of the deep ice cores that are inevitably drilled in holes filled with wall-retaining fluids [20]. It is therefore that the most critical step for determining reproducible and reliable Pb isotopes from such deep ice cores is to obtain the most central uncontaminated part of individual investigated core sections. Until now, the most successful decontamination method was mechanical decontamination of the outside of the core using sophisticated ultraclean procedures [21,22], which was originally developed by Clair Patterson and coworkers at the California Institute of Technology [23]. Apart from adequate decontamination procedures, the levels in the procedural blank should be sufficiently low within an acceptable range (<10% of the total Pb amount) to ensure accurate measurement of the Pb isotope ratios of ice core samples [22,24,25]. Due to difficulties in overcoming these challenges, reliable Pb isotope data from Antarctic deep ice cores have been obtained successfully only in a handful of laboratories [16,17].

Here, we present the first Pb isotope ratios in the EDC ice, dated from ~572 to 801 kyr B.P., corresponding to the two oldest glacial–interglacial cycles prior to the MBE. These data allow us to compare variations in Pb isotopic composition between our new data and those previously obtained for the recent two climate cycles in the same ice core [17], thereby contributing to evaluations of dust provenance changes potentially linked to latitudinal shifts of the SWW and the associated changes in atmospheric conditions between different climatic conditions before and after the MBE.

## 2. Materials and Methods

### 2.1. Ice Core Samples and Decontamination Procedure

We have analyzed 40 samples obtained from 30 core sections of the 3260-m EDC ice core, with depths from 2973.91 (572,800 yr B.P., MIS 15.1) to 3189.45 m (801,590 yr B.P., MIS 20.2) [26] (Figure 1). The depth and estimated age of each sample are given in Appendix A. Each of the 30 ice core sections (55 cm in length and 5 cm in radius) from the 3260-m EPICA Dome C ice core, drilled in a fluid-filled hole [1], was mechanically decontaminated using an acid-cleaned polyethylene lathe and ultraclean working procedures at the Korea Polar Research Institute (KOPRI) [20,27]. These involve the chiseling of successive layers of ice in progression from the contaminated outside toward the center of the section using acid-cleaned ultraclean stainless steel chisels. An external ~2 mm-thick layer of the most highly contaminated ice was scraped away before decontaminating. All the equipment used during the entire operation was extensively acid-cleaned prior to use, and the chiseling was performed inside a laminar flow class 100 clean bench located in a cold room at −15 °C. After the chiseling was completed, the inner core was then cut into two consecutive 20 cm long parts when the whole inner core was available. Altogether, 40 samples were analyzed for this study. Each sample was melted at room temperature in ultra-clean wide mouth low-density polyethylene (LDPE) 1 L bottles within a class 100 clean bench inside a class 10,000 clean room at KOPRI. About 10 mL aliquots were taken into acid-cleaned ultraclean 15 mL LDPE bottles and were then transported frozen to Curtin University in Perth, Australia, for Pb and Ba concentrations and Pb isotope analysis.

### 2.2. Mass Spectrometry

The procedures of sample processing and analysis by thermal ionization mass spectrometry (TIMS) have been described in detail by Burn et al. [25]. Briefly, the ice samples were melted and aliquots of ~1 to 10 g were transferred to PFA Teflon beakers, depending on the expected Pb contents based on the preliminary results of inductively coupled plasma sector field mass spectrometry (ICP-SFMS) analysis. A mixture of HNO_3_/HF/H_3_PO_4_ and a ^205^Pb/^137^Ba isotopically enriched spike solution were then added before the sample and tracer mixture were evaporated to dryness. The addition of the enriched isotopes ^205^Pb, ^137^Ba, and ^113^In enables the quantities of Pb, Ba, and In to be accurately determined by isotope dilution mass spectrometry (IDMS) [25]. The samples were loaded onto a degassed (4 A, 30 min), zone-refined rhenium filament (99.999% Re, 0.7 mm wide and 0.04 mm thick, H. Cross Company) with 4 µL of silica-gel and again evaporated to dryness. Prior to mounting the sample and silica-gel mixture, the filament was acid-cleaned using 7 µL of 1% Fisher “Optima” grade ultrapure HNO_3_ at 1.5 A. Samples were analyzed for Pb, Ba, and In concentrations using IDMS and Pb isotopes using a TRITON (Thermo Scientific) TIMS [25]. All ion beams were measured with a secondary electron multiplier (SEM), collecting ~300 isotope ratios per sample. The accuracy of the Pb, Ba and In concentrations is estimated to be ±10% (95% confidence interval), attributed mainly to the accuracy of dispensing the spike into the sample [25]. Two procedural blanks and two or more reference material samples containing a ~100 pg of NIST 981 SRM Pb isotopic standard were analyzed together with each batch of up to ~21 samples for quality control and monitoring of instrumental mass fractionation [25]. Procedural blanks amounted to be 0.4 ± 0.2 pg for Pb, 4.8 ± 1.4 pg for Ba and 1.3 ± 0.8 fg for In, accounting for ~12%, ~11% and ~9% at the lowest concentrations of Pb, Ba and In, respectively, that were measured in the samples. The concentration data in this study represent the blank-corrected values. Measurements of NIST 981 SRM showed that the instrumental isotopic fractionation was 0.11 ± 0.08% per atomic mass unit with precisions better than 0.1% for both ^206^Pb/^207^Pb and ^208^Pb/^207^Pb. The accuracy of the isotope ratio measurement was evaluated by analyzing the NIST 981 SRM. Very good agreements were obtained between the measured Pb isotope ratios and the certified values: 1.0945 ± 0.0004 versus 1.0933 for ^206^Pb/^207^Pb and 2.3657 ± 0.0017 versus 2.3704 for ^208^Pb/^207^Pb. The final isotopic ratios were obtained after correction for Pb blank contributions and their isotopic composition (1.171 ± 0.007 for ^206^Pb/^207^Pb and 2.430 ± 0.011 for ^208^Pb/^207^Pb), and instrumental mass fraction. Pb isotope ratio uncertainties at 95% confidence interval (Appendix A) are associated with the sample analysis, the isotopic composition of the Pb blank and the instrumental mass fractionation correction.

### 2.3. Validation Methodology for Decontamination Procedures

Although careful elimination of the significant contamination from the outside of the core sections were performed by mechanical chiseling as describe above, changes in the measured Pb and Ba concentrations and Pb isotope ratios as a function of radius from the outside to the inside of the core were investigated for the selected core sections to check the efficiency of the decontamination. Examples of such concentrations and isotope profiles are shown in Appendix A. In all cases, concentrations and isotope ratios are observed to level off at well-established plateau values in the central parts of the sections, indicating no transfer of the outside contamination to the inner part of the core.

### 2.4. An Isotope Mixing Model to Estimate Source Contributions to a Mixture

The relative contribution of individual potential sources to the observed isotopic composition in the EDC ice core samples was estimated using an isotope mixing model with the following equations:(1)(P 206bP 207b)m=∑i((P 206bP 207b)D,i×FD,i)+∑j((P 206bP 207b)V,j×FV,j)
(2)(P 208bP 207b)m=∑i((P 208bP 207b)D,i×FD,i)+∑j((P 208bP 207b)V,j×FV,j)
(3)$$\sum_{i}^{}\left(F_{D,i}\right)+\sum_{j}^{}\left(F_{V,j}\right)=1$$
where *m* indicates the measured sample isotopic abundances for mixtures of dust (*D*) and volcanic (*V*) end-members with different isotopic compositions and their fractional contributions (*F_D_* or *F_V_*). In order to identify the end-members contributing to the observed isotopic data, we examined several scenarios with different numbers and combinations of the potential end-members. Note that, when the number of end-members for individual dust (*i*) and volcanic (*j*) sources is greater than 3, the system of equations is considered underdetermined with fewer equations than unknowns and will have an infinite numbers of solutions. However, intrinsic conditions of 0 < *F* < 1 allow us to find more probable solutions even though more end-members are involved. We assumed that Ba is derived exclusively from dust sources with the upper crustal Pb/Ba ratio of 0.03^26^, which allows quantifying dust-derived (*∑F_D_*) and volcanic (*∑F_V_*) fractions of Pb contained in each sample:(4)$$\sum_{j}^{}\left(F_{D,j}\right)=\frac{0.03\times \left[Ba\right]}{\left[Pb\right]}$$

Since the constrained number of end-members helps to avoid meaningless and redundant solutions from the above equations, we divided the data into two separate groups of dust-dominant Pb (a dust-derived Pb fraction of >60%) and non-dust dominant Pb (a dust-derived Pb fraction of <60%) (see text). According to distinct isotopic signatures shown in Figure 2a, we then assumed that there are five potential source areas (PSAs) dominating dust in the EDC ice, i.e., Patagonia/TdF, the southern, middle, and northern central-western Argentina (S-CWA, M-CWA, and N-CWA, respectively), and the Puna-Altiplano Plateau (PAP) (see Appendix A). Note that we put Patagonia and TdF together as being the same source, because both isotopic fields overlap, making it difficult to distinguish from each other. Considering a wider range of Australian end-member values away from the measured Pb isotope ratios, we excluded the potential contribution of Australian dust (Figure 2a) [18]. As for the isotopic signatures of volcanic Pb, we assigned the McMurdo Volcanic Group (MVG) in Antarctica, the Central Volcanic Zone (CVZ) and the Southern Volcanic Zone/Austral Volcanic Zone (SVZ/AVZ) of the Andean Volcanic Arc (AVA), and Easter Island as the major volcanic sources (Figure 2b, Appendix A).

Proportions of the relative contribution of the selected end-members were determined by means of a Monte Carlo inversion approach consisting of two steps. First, the two-dimensional probability distribution of each end-member for the observed isotopic composition (^206^Pb/^207^Pb and ^208^Pb/^207^Pb) was derived by pooling literature data. The pooled probability distribution is equal to the normalized sum of two-dimensional uncertainty distribution of each datum. Then, the isotopic composition of individual selected end-members was randomly selected 10^6^ times from their respective probability distributions, so as to prepare 10^6^ sets of systems of equations in the pool. Second, the 10^6^ sets of systems of equations were solved for each sample. When a system of equations is underdetermined, we randomly selected more than 10^6^ solutions to focus on the reason of their distributions and statistics.

## 3. Results and Discussion

### 3.1. Elemental Concentrations and Pb Isotopes

The Pb and barium (Ba) concentrations and Pb isotopic compositions measured in the innermost parts of individual samples are illustrated in Figure 1, together with the profiles of dust flux or deuterium (δD) as a function of the age of the ice, and all data are listed in Appendix A. Figure 1 also shows published post-MBE data for the EDC ice core samples, dated from 2 kyr (MIS 1) to 220 kyr B.P. (MIS 7.3), by Vallelonga et al. [17]. Deuterium, an Antarctic temperature proxy [2], dust flux [5], and MIS numbers [28] are also given in Figure 1, facilitating comparison of data patterns along with different climatic stages.

Both the Pb and Ba concentrations, including published post-MBE data [17], show strong variability as a function of climate (δD) (Appendix A). Much higher concentrations of these elements are observed during cold periods between −450‰ and ~−430‰, with a rapid decline when δD values increase. A subsequent decrease in the concentrations appears in intermediate climatic stages with δD values from ~−430‰ to ~−410‰, and the concentration levels remain very low for δD values above ~−410‰. Note that the very high Pb concentration (108 pg/g) in the deepest ice at 3189.45 m (sample no. 40, ~801 kyr B.P., MIS 20.2) greatly exceeds the range of measured concentrations (<20 pg/g) when the δD values are below −430‰ (Figure 1 and Appendix A). Despite the stratigraphic continuity of multi-parametric climatic records above 3200 m as described previously [2], the very high Pb content may suggest non-climate-related influences, possibly bedrock, and thus the sample is not included when interpreting climate signals. The Pb concentrations are positively well correlated with Ba (a conservative crustal reference element) during both pre-MBE and post-MBE cold and intermediate climatic periods with significant Spearman’s correlation coefficients between 0.697 and 0.850 at *p* = 0.05 or *p* = 0.01, respectively. This reflects that the changes in the Pb concentrations during these different climatic periods are primarily dependent on the dust concentrations. There is a lack of significant Spearman’s correlation during pre-MBE interglacials, while the correlation between the two elements appears to be strong (0.765 at *p* = 0.01) during post-MBE interglacials, most likely associated with an enhanced en route wet removal of volcanic Pb derived from extra-Antarctic volcanoes as discussed later in this paper.

Aside from Pb and Ba, the concentrations of indium (In) were also measured in our samples (Appendix A), which are the first data ever obtained for the EDC ice. The In concentrations, ranging from 2.15 to 117 fg/g, show a general pattern of variability, with higher levels during glacial maxima with δD values below ~−430‰ and lower levels during warm climates, except for some deviations (for example, sample nos. 20, 27, and 29) from climate-related concentration levels, coinciding with moderately to highly elevated Pb concentrations (Appendix A). These deviations can be attributed primarily to increased volcanic fallout of both In and Pb, which are elements sensitive to volcanic emissions [16,17,29,30].

The Pb isotope ratios vary from 1.1824 to 1.2332 for ^206^Pb/^207^Pb and from 2.4556 to 2.4939 for ^208^Pb/^207^Pb with mean values of 1.2022 and 2.4715, respectively (Appendix A). Temporal changes in the pre-MBE ^206^Pb/^207^Pb ratios are less well correlated with temperature in Antarctica, while the post-MBE values during 2–220 kyr B.P. show a general trend associated with climatic conditions with higher values during warm or less cold periods and lower values during very cold climatic stages (Figure 1). This difference may be partly due to larger age intervals (~90–250 years) integrated by individual pre-MBE samples relative to the post-MBE ones (~10–60 years), resulting in the smoothing of pre-MBE source-specific Pb isotopic signatures. Note that long-term in situ processes have resulted in the aggregation of mineral particles and a rearrangement of ionic impurities at depths below ~2900 m in the EDC ice core [5,31]. However, we exclude the impact of such alterations on temporal changes recorded in our data, because the stratigraphic continuity of climate-related changes in elemental concentrations is well observed (Appendix A) and the isotopic compositions of Pb are not significantly affected by secondary physical or chemical fractionation processes [32].

### 3.2. Comparison of Dust-Derived Pb Isotopic Compositions before and after the MBE

By combining the δD values and their relationships with our data, we divided Antarctic climates into three climatic conditions, i.e., cold glacial (δD < −430‰), intermediate glacial (−430‰ < δD < −410‰), and warm interglacial (δD > −410‰) stages, facilitating a comparison of climate-related changes in our data. Recently, Gili et al. [18] constrained well-defined potential source areas (PSAs) of dust in southern South America (SSA) by coupling Pb isotopic compositions in the EDC samples from Vallelonga et al. [17] with new Pb isotopic data from unexplored PSAs in SSA. In the study of Gili et al. [18], however, Pb isotopic data by Vallelonga et al. [17] were categorized only into cold (δD < −435‰) and warm (δD > −435‰) climatic stages for individual samples, thereby hampering the understanding of progressive changes in the provenance of dust in response to a changing climate. Another perspective to consider is that natural Pb in Antarctic ice could originate from both dust and volcanoes [16,17]. An estimate of the dust contribution was made based on the Pb/Ba ratio of upper continental crust, ~0.03 [17,33]. According to the fraction of Pb of dust origin, isotopic signatures in each sample were divided into dust-dominant Pb (a dust-derived Pb fraction of >60% with Pb/Ba < 0.05) and non-dust dominant Pb (a dust-derived Pb fraction of <60% with Pb/Ba > 0.05), as done for post-MBE isotopic signatures [17]. A smaller estimated dust contribution indicates an increased non-dust (that is, volcanic) contribution.

In Figure 2a, both the pre-MBE and post-MBE dust-dominant Pb isotopic compositions, combined with the PSAs fields defined by Gili et al. [18], exhibit a significant variability between different climatic stages. A larger number of isotopic signatures of dust-dominant Pb during post-MBE cold climatic conditions tend to distribute along the edge of a mixing isotopic area among the PSAs of dust in SSA, Patagonia, Tierra del Fuego (TdF), and the southern and middle central-western Argentina (S-CWA and M-CWA, respectively) (Appendix A), generally displaying a substantial variation with respect to ^208^Pb/^207^Pb, with several data points shifting towards the isotopic field of the Puna-Altiplano Plateau (PAP) in the Andean Cordillera. For comparison, the pre-MBE cold glacial isotopic compositions are within a small range between the fields of Patagonia, TdF, S-CWA and M-CWA (Figure 2a).

A different situation emerges for the isotopic compositions during intermediate and warm climatic conditions before and after the MBE. The post-MBE isotopic signatures during intermediate climatic stages are characterized by a significant variation in the ratios of ^206^Pb/^207^Pb, moving to the M-CWA and northern CWA (N-CWA) fields, excluding two data points within the southeastern Australian field (Figure 2a). The pre-MBE isotopes remain within a small range as in the cold climates with the exclusion of sample no. 27 (Figure 2a), for which all elemental concentrations are relatively high compared to those observed during the corresponding climatic conditions (Appendix A), reflecting a strong influence of volcanic emissions. Subsequently, the post-MBE warm interglacial isotopes vary over a wider range of the mixing line between Patagonia and the McMurdo Volcanic Group (MVG) in northern Victoria Land, Antarctica. The pre-MBE interglacial data also fall on a post-MBE compositional area. The Pb isotopic signatures in the EDC ice moving towards the most radiogenic composition have been attributed to increased volcanic contributions from the MVG [17,18]. However, the dust fractions of the corresponding samples do not show any systematic changes with changing climate before and after the MBE (Appendix A), suggesting that the occurrence of climate-related isotopic variations in dust-dominant samples could be primarily due to changes in the main dust sources.

### 3.3. Dust Provenance and Its Relevance to a Shift of the SWW

We made a rough estimate of the proportional individual dust source contribution to the observed isotopic mixture using an isotope mixing model (see Methods). This approach provides fundamental insights into a quantitative comparison of climate-related dust provenance changes between the pre-MBE and post-MBE, albeit with a small pre-MBE sample size (Appendix A). As the results from the Monte-Carlo approaches were not normally distributed, we used median values instead of means to reduce the bias in estimating the relative contributions of individual sources. Note that Patagonia and TdF are considered as being the same source, because both isotopic fields overlap, making them difficult to distinguish from each other (Figure 2a).

The model simulation results show the importance of Patagonia/TdF as a dust contributor during cold and intermediate glacial climates before and after the MBE (Figure 3a and Appendix A). Compared to previous studies that identified Patagonia/TdF as the main sources for central East Antarctic dust during very cold glacials [8,9,10,11,12,14,15], it is interesting to observe that the dominance of Patagonia/TdF dust remain in both the pre-MBE and post-MBE intermediate glacials, despite a significant reduction of dust fluxes during these periods (Figure 1). This suggests the persistence of mechanisms responsible for production and transport of Patagonia/TdF dust to the EAP under both cold and intermediate glacial climates before and after the MBE. The dominance of Patagonia/TdF dust during these glacial stages would be most likely attributed to the northward expansion and/or intensification of the SWW [6,15,47], combined with the glacial advances of the Northern and Southern Patagonian Ice Sheet (PIS) [48,49], stretched from 37 to 56°S, and the associated increase of glaciofluvial outwash deposits, allowing the enhanced dust entrainment associated with an increase in the vigor of atmospheric circulation [12,48,49]. In addition, the persistent Patagonia/TdF glacial dust would be attributed in part to a lowered sea level extending the Argentine continental shelf area [12,50], when fine-grained shelf sediments were delivered from Patagonia [49], which in turn retained a dominant Patagonian dust signature [51].

Apart from the major dust contribution from Patagonia/TdF, the CWA emerges as an important dust supplier to the EAP during both the pre-MBE and post-MBE cold and intermediate glacial stages (Figure 3a and Appendix A), as in the Sr-Nd isotopic constraints on the CWA origin of dust in recent glacial periods [15]. This suggests a northward extension of the SWW belt to the CWA (27–39°S), far north of its present position of the strong zonal winds between ~45 and ~60°S [52], during these glacial stages. Although an equatorward displacement and strengthening of the SWW belt during glacial periods remains under debate (e.g., Kohfeld et al. [53] and references therein), our data support the hypotheses that during the glacial conditions, an equatorward shift in the SWW was as large as 7–10° relative to its interglacial position [6] and the strengthening of the northern margin of the SWW occurred at 33–40°S [47]. The northward shift of the northern edge of the SWW belt would have induced more vigorous northwesterly winds and the consequent increase in dust emissions over CWA [15], coupled with drier conditions in the SSA north of 40°S [53,54], enhancing the input of glacial dust from these areas to the EAP. Subsequently, our model simulated the absence of a significant PAP dust contribution during both the pre-MBE and post-MBE cold and intermediate glacial stages (Figure 3a and Appendix A). This situation is inconsistent with the hypothesis of its potential contribution during the post-MBE glacials, attributed to an equatorward movement of the subtropical westerly jet stream (SJT) over the PAP, a high elevation basin (~4000 m a.s.l.) [14,15,18,55]. Further research is needed to examine conflicting perspectives on the contribution of PAP to the glacial dust in the EAP.

Finally, a new feature that has never been previously noticed is the emergence of CWA as the important dust source, accounting for an average contribution of ~42%, during post-MBE interglacials, along with a substantial reduction in the Patagonia/TdF contribution (11 ± 32%) (Figure 3a and Appendix A). Comparatively, Patagonia/TdF was the largest supplier (58 ± 30%), followed by the CWA (~24%), during pre-MBE interglacials (Figure 3a and Appendix A). Based on the difference in the average δD values for dust-dominant samples between the pre-MBE and post-MBE interglacials (−405 ± 1‰ and −399 ± 7‰, respectively) (Appendix A), there may be a critical climatic threshold of δD value between −399 and −405‰, beyond which the transfer of dust from Patagonia/TdF to the EAP have been greatly reduced, allowing an increase in the relative magnitude of the CWA contribution in line with a significant reduction of dust fluxes in the EAP. A similar threshold mechanism was observed for the non-sea-salt calcium (nssCa^2+^) flux, South American dust proxy, and δD in the EDC ice [56], which show a fairly close relationship between these two proxies during glacial periods up to δD value of ~−402‰, followed by disappearance of such correlation beyond this δD value, reaching interglacial nssCa^2+^ flux level [56]. The threshold mechanisms observed in our data and nssCa^2+^ flux would be related to the SWW that has moved further poleward during warmer interglacials, potentially causing subsequent changes in wind strength in the source areas and precipitation patterns in the Southern Ocean [50,56,57].

Besides a 7–10° shift of the SWW at glacial-interglacial timescales [6], changes in the SWW occurred in sensitive response to changes in climatic conditions over decadal to multi-millennial timescales [58,59,60]. This resulted in a northward displacement and/or stronger wind intensity at mid-latitudes between ~40 and 50°S under cooler climates and the opposite behavior during warmer climates [60,61]. According to a poleward shift of the SWW under warming climates, precipitation patterns move towards higher latitudes [62], which, in turn, increase precipitation [5,53,54] in the Southern Ocean where dust is transported out of Patagonia/TdF to the EAP [63], hence enhancing a dust wet removal from the atmosphere [5,57]. We thus attribute the modified hydrological cycle, combined with a decrease in dust production and entrainment in Patagonia/TdF [56], to a small fraction of dust from Patagonia/TdF during warmer post-MBE interglacials. Conversely, the persistent transfer of dust from Patagonia/TdF to the EAP during pre-MBE interglacials reflects a weakened precipitation over the Southern Ocean, as the SWW moves northward with the cooler interglacial climates [60,61]. A northward displacement of the SWW is also associated with an enhanced dust supply and strong wind uptake as a consequence of more frequent cyclonic influence and more steady zonal winds in SSA as in the case for winter conditions in the current climate [63]. Interestingly, the EDC ice core record shows a substantial increase of dust fluxes during cooler pre-MBE interglacials (MIS 15.1, 15.5, 17.3 and 19.3), compared to the warmer post-MBE interglacials (MIS 1 and 5.5) [3,5]: the average (±SD) dust flux during the pre-MBE MIS 15.1, 15.5, 17.3 and 19.3 interglacials is 0.85 ± 0.61 mg/m^2^/yr, which is ~2 times the average flux (0.45 ± 0.20 mg/m^2^/yr) of the post-MBE MIS 1 and 5.5 interglacials. This difference is also shown in dust fluxes calculated in our samples corresponding to pre-MBE and post-MBE interglacial periods (Appendix A). Furthermore, the EDC interglacial ice CO_2_ levels prior to the MBE were ~30–40 ppm lower than those after the MBE [64,65]. This may be partly related to the weakened strength of the Antarctic Circumpolar Current in response to a northward shift in the mean position of the SWW, resulting in reduced ventilation of respired CO_2_ in the deep ocean to the atmosphere [6,7]. However, any speculation must be treated cautiously due to the lack of a comprehensive understanding of processes and mechanisms for the lower interglacial CO_2_ levels prior to the MBE [66].

### 3.4. Volcanic Isotopic Signatures and Their Atmospheric Implications

An interesting difference in the Pb isotopic signatures between the pre-MBE and post-MBE periods is observed when the ^206^Pb/^207^Pb ratios are plotted as a function of relative dust-derived Pb proportions (Figure 4). The ^206^Pb/^207^Pb ratios are relatively constant and less radiogenic when dust-derived Pb fractions are above ~60%, but both datasets display a diverging pattern in concert with increasing volcanic contributions. The post-MBE non-dust dominant ratios show a clear compositional trend moving toward the most radiogenic MVG volcanic field (Figure 4), which was interpreted as a result of increasing amount of volcanic Pb from the MVG [17,18]. A three-isotope plot (^206^Pb/^207^Pb versus ^206^Pb/^207^Pb) of the post-MBE non-dust dominant samples also exhibits a well-defined approach to the MVG field in line with a significant reduction from group C to group A in the mean proportions of dust-derived Pb (Figure 2b), supporting the volcanic signals associated with the radiogenic end-member of the MVG.

On the other hand, the pre-MBE ^206^Pb/^207^Pb non-dust dominant ratios show no distinct increasing trend (Figure 4). The pre-MBE isotopic compositions in Figure 2b are mostly plotted within a relatively small range between the Andean Volcanic Arc (AVA), suggesting a strong influence of volcanic Pb derived from extra-Antarctic volcanoes, particularly the AVA. The isotopic compositions of sample nos. 20 and 29 show a shift towards a more radiogenic MVG field (Figure 2b). Combining with their isotopic constraints, very high In concentrations and In/Pb ratios (>0.02) larger than a typical upper continental crustal value (~0.004) [33] in these samples (Appendix A and Appendix A) reflect the significant sources of volcanic emissions probably from the Mt. Erebus, the world’s southernmost active volcano, where the Erebus plume is enriched in In relative to Pb [29].

Our model simulations represent pronounced differences in the magnitude of the associated volcanic source contributions under different climatic conditions before and after the MBE. A remarkable difference between the pre-MBE and post-MBE is a substantially smaller contribution from extra-Antarctic volcanoes during warmer post-MBE interglacials relative to cooler pre-MBE ones (Figure 3b and Appendix A). This would be due to the enhanced en route wet removal resulting from a more intense hydrological cycle over the South Ocean under warmer post-MBE climatic conditions [5,53,54], as considered to be an important parameter for a significant reduction in the dust transport from Patagonia/TdF to the EAP during warmer interglacials after the MBE. This hypothesis, however, contradicts a previous study proposing that high super-chondritic (volcanic) platinum (Pt) and iridium (Ir) fluxes in the EAP during the post-MBE interglacials (MIS 1 and 5) were probably due to the enhanced advection of air masses from lower latitudes to the EAP according to a weakened polar vortex [68]. This contradiction may arise from differences in the removal efficiency of individual volcanogenic metals, mostly abundant in the aerosol fraction of the plume as halides, sulfates, sulfides, and/or metals [69,70], from the atmosphere primarily via wet deposition processes including within-cloud and below-cloud scavenging. The modeling study of volcanic emissions showed that volcanic PbCl_2_, a highly soluble species enriched in degassing volcanic emissions, decreased exponentially in the atmosphere with the distance from the volcano due to the rapid wet deposition [71]. In contrast, refractory and chemically inert Ir and Pt that are enriched in volcanic emissions in association with the high fluorine content, compose water insoluble fluoride species (e.g., iridium hexafluoride, IrF_6_) [72], which would have the potential for further atmospheric transport over long distances. Previous studies identified well-defined Pt and Ir peaks in the volcanic layers of Antarctic and Greenland snow deposits without Pb enrichments coincident with these peaks [73,74], supporting our hypothesis. As a result, we can infer that extra-Antarctic volcanic Pb signatures in the pre-MBE interglacial climates are most likely related to a longer lifetime for volcanic aerosols from quiescently degassing volcanoes, associated with the reduction of wet removal efficiency in response to cooler pre-MBE interglacial climates. Together with a prolonged lifetime of volcanic aerosols, a northward shift and/or extension of the SWW belt during cooler pre-MBE interglacials could have enhanced an advection of volcanic Pb from volcanoes outside Antarctica to the EAP. Our results point to a substantial weakening of the hydrologic cycle in line with cooler sea surface temperature of the Southern Ocean during pre-MBE lukewarm interglacials relative to the post-MBE interglacials [75,76].

The last notable difference is that the MVG contribution is estimated to be dominant during post-MBE cold and intermediate periods, while it remained relatively small whatever the pre-MBE periods (Appendix A). The dominant MVG contribution is likely due to a stronger dynamical isolation of the polar vortex area over Antarctica under these climate conditions [11], reducing the advection of Pb emitted from quiescent degassing of extra-Antarctic volcanoes to the EAP. Given the above, we can infer that the MVG contribution might have increased in the cold or intermediate climatic stages prior to the MBE. However, our data do not support such a speculation (Figure 3b and Appendix A). One possible explanation could be linked to the evolution of the MVG volcanoes, particularly Mt. Erebus (Appendix A), which was estimated to be the most potential source within Antarctica for the volcano-derived trace elements in Antarctic ice [29,77]. The evolution of Erebus volcano is divided into the following phases: (1) a proto-Erebus shield building phase (1.3 Ma–1.0 Ma); (2) a proto-Erebus cone building phase (1.0 Ma–250 ka); and (3) the modern-Erebus cone building phase (250 ka–present) [78,79]. During the last phase, the bulk of the present-day Erebus edifice, characterized by periodic Strombolian eruptions from a persistent lava lake, was formed in connection with increased volcanic activity and a large-volume lava extrusion [78,79]. Although hypothetical, we thus suggest that no significant contribution of volcanic Pb from the MVG prior to the MBE would primarily be related to the evolution of Erebus volcano, with the potential contribution from intermittent volcanic activity of Erebus before the MBE as seen for sample nos. of 20 and 29 (Figure 2b).

## 4. Conclusions

New Pb isotope data for the oldest part of the EDC ice core, combined with published post-MBE data, provide the first evidence for Patagonia/TdF and CWA as the persistent sources of East Antarctic dust during both cold and intermediate glacial climates before and after the MBE. The observed isotopic signatures, coupled with isotope mixing model results, suggest the emergence of CWA as the largest dust supplier to the EAP, along with a weakened dust input from Patagonia/TdF, during warmer post-MBE interglacials, which has never been noticed before. Conversely, the dominance of Patagonia/TdF sources persisted during cooler pre-MBE interglacials. Our results also represent a large difference in the potential sources of volcanic-associated Pb deposits between the pre-MBE and post-MBE interglacials, with a small fraction of volcanic Pb transferred from extra-Antarctic volcanoes during warmer post-MBE interglacials, contrary to abundant transfer prior to the MBE. A difference in the average values of δD in both interglacial data sets suggests that there is a critical climatic threshold of Antarctic temperature, corresponding to a δD value between −399 and −405‰, beyond which the transfers of both dust and volcanic Pb from Patagonia/TdF and extra-Antarctic volcanoes, respectively, to the EAP have been significantly reduced. We infer that such climate-related differences are most likely attributed to a poleward shift of the SWW in combination with a stronger hydrologic cycle in the Southern Ocean as a result of warmer climatic conditions surpassing the climatic threshold, thereby enhancing the en route wet removal of aerosols from the atmosphere, and vice versa during cooler pre-MBE interglacial climates below the threshold. Our findings highlight sensitive responses of the SWW belt and the associated southern climatic system to stepwise Antarctic warming under the interglacial conditions. Due to constraints in conducting this study with a relatively small sample size, however, it will be necessary to obtain a high-resolution record of changes in Pb isotopic composition prior to the MBE, thus helping to validate our findings.

## Figures and Tables

**Figure 1 molecules-27-04208-f001:**
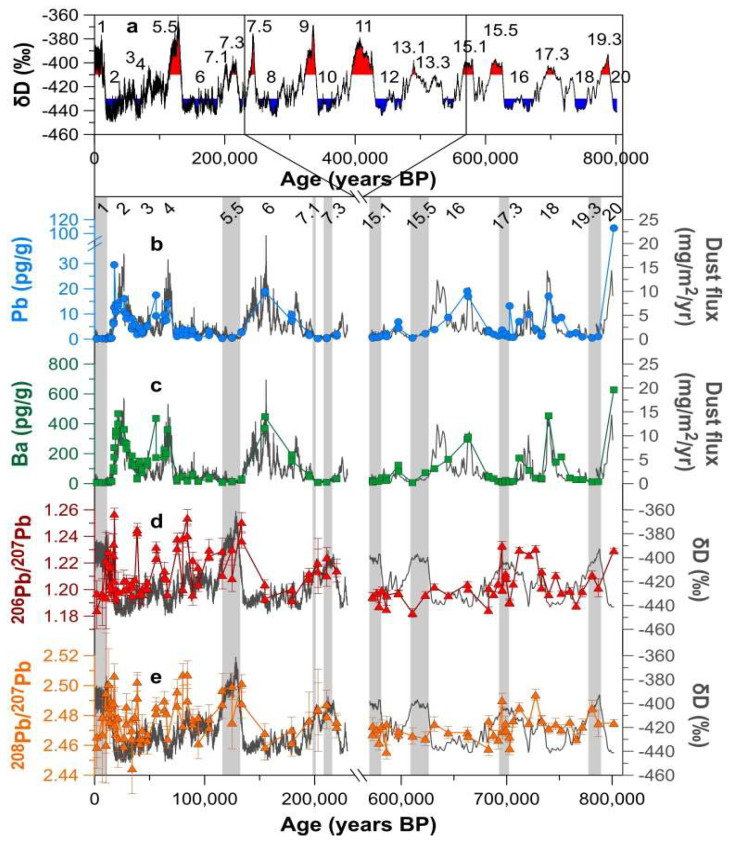
Changes in Pb and Ba concentrations, and Pb isotopic compositions from the EDC ice core. Also included are previously published data from the EDC ice core during the reporting period from 2 kyr (MIS 1) to 220 kyr B.P. (MIS 7.3) [17]. (**a**) The full EDC δD (Antarctic temperature proxy) profile [2] with Marine Isotope Stage (MIS) numbers [28]. (**b**,**c**) Changes in Pb and Ba concentrations and their deposition fluxes. Flux of dust in the EDC ice core [5] is shown as a gray solid line for comparison. (**d**,**e**) Changes in ^206^Pb/^207^Pb and ^208^Pb/^207^Pb ratios. The δD record in the EDC ice core is shown as a gray solid line for comparison. The vertical grey bars represent the interglacial periods when the δD values are above −405‰. All uncertainties for ^206^Pb/^207^Pb and ^208^Pb/^207^Pb ratios are 95% confidence intervals.

**Figure 2 molecules-27-04208-f002:**
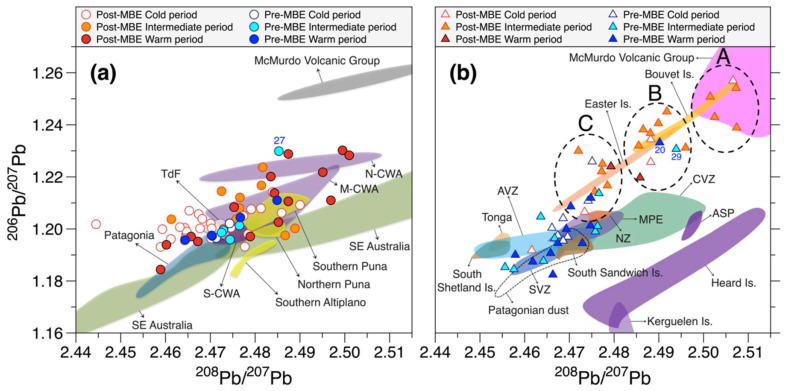
A plot of ^206^Pb/^207^Pb versus ^208^Pb/^207^Pb in the EDC ice core. Also included are post-MBE data previously reported from the EDC ice core [17]. The observed data are divided into isotopic compositions for (**a**) dust-dominant samples with a dust-derived Pb fraction of >60% and (**b**) non-dust dominant samples with a dust-derive Pb fraction of <60% (see text). The isotopic fields of the potential sources of dust in (**a**) and volcanic Pb in (**b**) were derived from the literature: Puna-Altiplano Plateau (PAP) [18], central-western Argentina (CWA) [18], Patagonia/Tierra del Fuego (TdF) [18], southeastern (SE) Australia (Murray-Darling Basin) [17,34], McMurdo Volcanic Group (McM) [35,36], South Sandwich Island [37], South Shetland Island [38], Bouvet Island [39], New Zealand (NZ) [40], Easter Island and Tonga [39], Marion and Prince Edward (MPE) [41], Kerguelen Island [42], Heard Island [43], Amsterdam-St. Paul Island (ASP) [44], SVZ: Southern Volcanic Zone, and AVZ: Austral Volcanic Zone of the Andean Volcanic Belt [45,46]. Group A, B, and C within the dashed circles in Figure 2b exhibit stepwise changes in the post-MBE isotopic compositions to more radiogenic values as progressive reduction in the mean proportions of dust-derived Pb (29 ± 13% for group A, 42 ± 11% for group B, and 51 ± 6% for group C). Sample numbers mentioned in the text are also shown in Figure 2. The acronyms next to the fields are defined in Appendix A.

**Figure 3 molecules-27-04208-f003:**
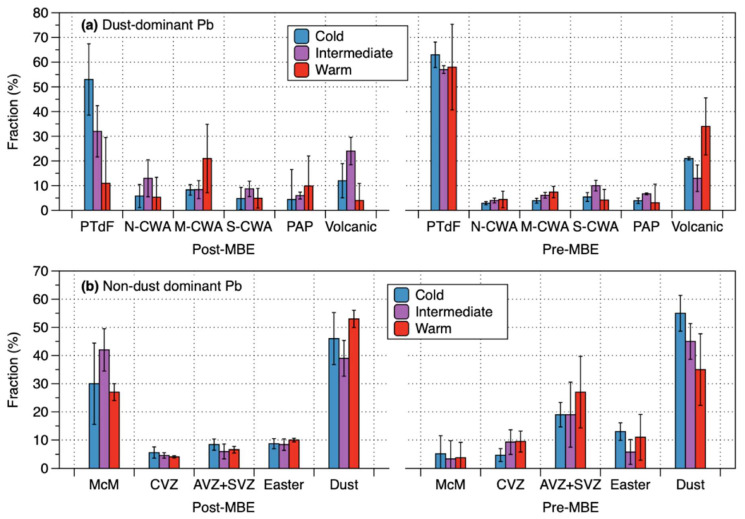
The relative contributions (in %) of (**a**) the potential dust sources for dust-dominant samples and (**b**) the potential volcanic sources for non-dust dominant samples calculated using an isotope mixing model (see Methods and Appendix A). The acronyms are defined in Appendix A. The bar graph shows the median values and vertical lines represent standard deviations of a data set.

**Figure 4 molecules-27-04208-f004:**
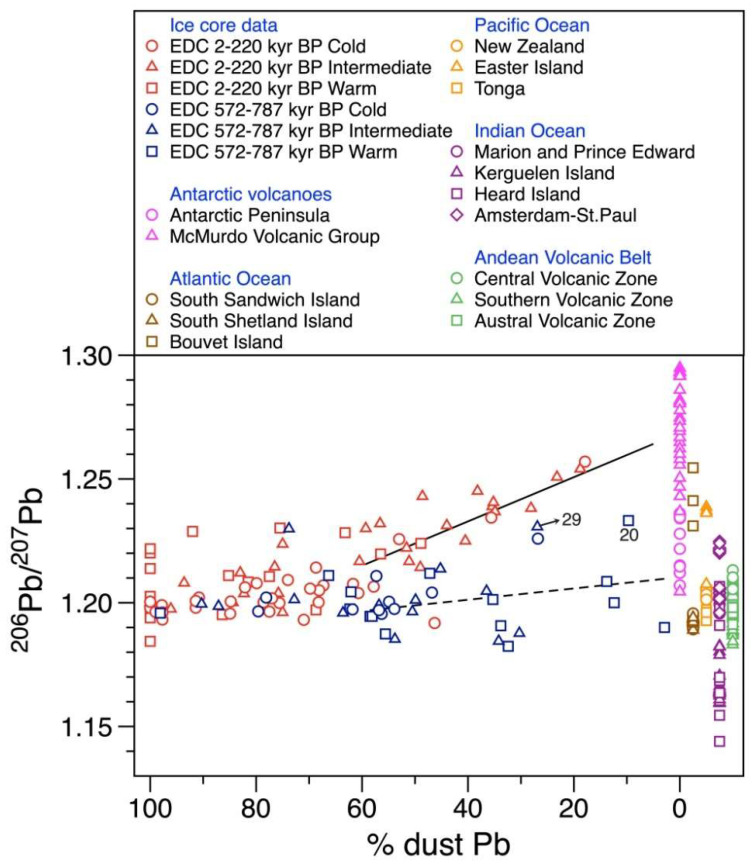
Comparison of ^206^Pb/^207^Pb ratios between the pre-MBE and post-MBE intervals as a function of the dust fraction of Pb in the EDC ice core samples. All uncertainties are 95% confidence intervals. The end-members of volcanic ^206^Pb/^207^Pb ratios for the potential volcanic sources come from published literature: Antarctic Peninsula basalts [67], McMurdo Volcanic Group [35,36], South Sandwich Island [37], South Shetland Island [38], Bouvet Island [39], New Zealand [40], Easter Island and Tonga [39], Marion and Prince Edward [41], Kerguelen Island [42], Heard Island [43], Amsterdam-St. Paul Island [44], and Andean Volcanic Belt [45,46]. The locations of individual volcanoes are shown in Appendix A. Note that the fraction of dust Pb in excess of 100% shown in part of the post-MBE samples [17] was considered to be 100%. The least squares lines are shown for the post-MBE (solid line) (Spearman’s correlation of 0.736 at *p* < 0.01) and pre-MBE (dashed line) isotopic data (no correlation with Spearman’s correlation of 0.207) with a dust-derived Pb fraction of <60%. Sample numbers mentioned in the text are also shown in Figure 4.

## Data Availability

Data presented in this study are available in Appendix A and will be archived at the World Data Center PANGAEA (https://www.pangaea.de) (accessed on 29 June 2022).

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
