# Peer review of "Lead Isotopic Constraints on the Provenance of Antarctic Dust and Atmospheric Circulation Patterns Prior to the Mid-Brunhes Event (~430 kyr ago)"

_molecules, 2022, doi:10.3390/molecules27134208_

Round 1
Reviewer 1 Report
I do not see any weaknesses of this work From the point of view of chemical analysis and the correctness of the applied research methodology. I propose to correct the title of subchapter 2.2. to Inductively coupled plasma mass spectrometry" and complete the results described in this chapter with with method detection limit for Pb isotopes.
Author Response
Reviewer 1
I do not see any weaknesses of this work from the point of view of chemical analysis and the correctness of the applied research methodology. I propose to correct the title of subchapter 2.2. to Inductively coupled plasma mass spectrometry" and complete the results described in this chapter with method detection limit for Pb isotopes.
Response: All of the data used in this study were determined using TIMS. ICP-MS was just used to take a look at the concentration levels of Pb, which helped determine the sample volume for the TIMS analysis. Thus, we’d like to leave the title of subchapter 2.2 as it is. As for the detection limit for Pb isotopes, the more important points are the procedural blank level for Pb, precision and accuracy. As noted in section 2.2, the procedural blank of Pb was 0.4 ± 0.2 pg, which is negligible when considering the total amount of Pb (50-100 pg) in the samples used for our analysis. Despite that, the measured Pb isotopes were corrected with Pb blank contributions and their isotopic composition as described in section 2.2. Both analytical precision and accuracy for Pb isotope measurement are also given in section 2.2.
Reviewer 2 Report
The manuscript by Han et al. presents new Pb isotope data for the EDC ice core (oldest part), which is a nice supplement to previous core data. Mixing modelling of the dataset assessed the dust sources in central East Antarctic prior to the MBE and addresses the impacts of atmospheric circulation patterns, highlighting the responses of the atmospheric conditions to stepwise Antarctic warming under the interglacial conditions. While this work is interesting and in good shape, some issues should be addresses before publication. My comments are presented below.
Major comments
Ø The research gap in this field and the significance of this work should be better highlighted in the Introduction. The rationale for lead isotope tracing of dust sources and their advantages over other tracers should also be better explained.
Ø Section 3.1 and Figure 1: The information represented by each parameter should be better explained.
Ø As the discussions relied much on the source apportionment, the endmembers should be better constrained.
Ø It seems to me that some of the discussions are too long and somewhat speculate, which could be simplified.
Specific comments
Line 20: The full name of EPICA should be provided for the first time it appears in the abstract.
Lines 75-81: The differences between this and the existing studies and the significance of this work should be clarified.
Lines 154 and155: What do i and j mean?
Lines 154-155: these lines are confusing to me…the equations are probably redundant or need to be revised since five endmembers are included in the following text.
Lines 169-172: I am not sure dividing the data into two separate groups can help to avoid meaningless and redundant solutions.
Lines 172-180:The endmembers should be better constrained or described as they largely determine the source apportionment results.
Line 182 Does the distribution of model results normally distributed?
Figure 2 and Figure 4: Suggest to move the sources of the endmembers from SM to the main text.
Author Response
Reviewer 2
The manuscript by Han et al. presents new Pb isotope data for the EDC ice core (oldest part), which is a nice supplement to previous core data. Mixing modelling of the dataset assessed the dust sources in central East Antarctic prior to the MBE and addresses the impacts of atmospheric circulation patterns, highlighting the responses of the atmospheric conditions to stepwise Antarctic warming under the interglacial conditions. While this work is interesting and in good shape, some issues should be addresses before publication. My comments are presented below.
Major comments
- The research gap in this field and the significance of this work should be better highlighted in the Introduction. The rationale for lead isotope tracing of dust sources and their advantages over other tracers should also be better explained.
Response: Accordingly, we have added some sentences in lines 61-72 in Introduction to emphasize the significance of our work.
- Section 3.1 and Figure 1: The information represented by each parameter should be better explained.
Response: We have inserted a sentence in lines 216-218, saying “Deuterium, an Antarctic temperature proxy [2], dust flux [5], and MIS numbers [28] are also given in Figure 1, facilitating comparison of data patterns along with different climatic stages”, accordingly.
- As the discussions relied much on the source apportionment, the endmembers should be better constrained.
Response: According to this comment and the comment below, we have moved the references for individual endmembers from Figures S2 and S4 in Supplementary Materials to Figures 2 and 4. And we have inserted a sentence of “Considering a wider range of end-member values away from the measured Pb isotopic values, we excluded the potential contribution of Australian dust (Figure 2a)” in lines 191-193 in section 2.4.
- It seems to me that some of the discussions are too long and somewhat speculate, which could be simplified.
Response: We agree with this comment that some of the discussions are too long and somewhat speculate. This may be mainly due to a small sample size as described in the text. As we noted in conclusion, our findings need to be validated by further research based on a larger sample size. Instead, we have removed sentences concerning additional input of dust from PAP during the pre-MBE glacials. This description was based on the mean values of model calculation. However, when we use the median values according to the comment given by another reviewer, the situation was changed for the PAP dust contribution. Therefore, we removed sentences (lines 386 and 391-403 in the original manuscript) related to the hypothesis given to explain the difference in the PAP contributions between pre-MBE and post-MBE. This would partly follow this comment.
Specific comments
- Line 20: The full name of EPICA should be provided for the first time it appears in the abstract.
Response: We have inserted the full name of EPICA accordingly.
- Lines 75-81: The differences between this and the existing studies and the significance of this work should be clarified.
Response: We have added some sentences in Introduction to emphasize the significance of our work according to the comment.
- Lines 154 and155: What do i and j mean?
Response: i and j are the number of end-members for individual dust and volcanic sources, respectively, used for model calculation. We have inserted i and j in the text, saying “when the number of end-members for individual dust (i) and volcanic (j) sources is greater than 3”.
- Lines 154-155: these lines are confusing to me…the equations are probably redundant or need to be revised since five endmembers are included in the following text.
Response: We divided the end-members into two groups (dust and volcanic) in order to use the equation (5) as an additional constraint for our mixing model, which allowed us to involve more than 3 potential end-members.
- Lines 169-172: I am not sure dividing the data into two separate groups can help to avoid meaningless and redundant solutions.
Response: This is because the measured Pb isotope compositions display a binary mixture of aerosols contributed from two major Pb sources, that is, dust and volcanic emissions as described in the text. We provided insights and interpretation in the text primarily based on the proportional source contributions of dust or volcanic sources obtained from an isotope mixing model using two separate groups.
- Lines 172-180: The endmembers should be better constrained or described as they largely determine the source apportionment results.
Response: Please see response to the above comment 3. In addition, we have inserted an additional sentence in lines 191-193 in the revised manuscript.
- Line 182 Does the distribution of model results normally distributed?
Response: The results from the Monte-Carlo approaches were not normally distributed. This was also commented by another reviewer. Thus, we clarified this situation, inserting a sentence “Because the results from the Monte-Carlo approaches were not normally distributed, we estimated the relative contributions of individual sources using median values instead of means to minimize bias’ in lines 341-343 in section 3.3. Then we modified the text according to this change. We have replaced the mean values with the medians in Table S3 and S4. Figure 3 was also modified with the median values. In fact, no large differences between mean and median values are observed, except for the contribution from PAP dust during the post-MBE cold periods, for which the estimated contribution during the post-MBE cold periods decreased from mean value of 15 ± 21% to median value of 4.4 ± 21%. According to this situation, we have removed the sentences related to this situation (lines 386 and 391-404 in the original manuscript).
Figure 2 and Figure 4: Suggest to move the sources of the end-members from SM to the main text.
Response: We have moved the references for individual end-members from Figures S2 and S4 in Supplementary Materials to the captions of Figures 2 and 4 accordingly.
Reviewer 3 Report
This is the review of the manuscript molecules-1773281 entitled “Lead isotopic constraints on the provenance of Antarctic dust and atmospheric circulation patterns prior to the Mid-Brunhes Event (~430 kyr ago)”. In this study, the authors perform Pb isotope analysis in the EDC ice core, in order to evaluate the dust provenance changes and the associated changes in atmospheric conditions between different climatic conditions before and after the MBE. The manuscript is well written and can represent an important contribution to the scientific literature. However, I believe this manuscript should be published after a minor revision.
Minor Revision:
The results of the mixing model that the authors use, perform large standard deviations compared with the mean values. Even though these large deviations are statistical acceptable, lead to a bad data description. Therefore, I would suggest presenting the median values of their samples and the range for better description of their results.
Author Response
Reviewer 3
This is the review of the manuscript molecules-1773281 entitled “Lead isotopic constraints on the provenance of Antarctic dust and atmospheric circulation patterns prior to the Mid-Brunhes Event (~430 kyr ago)”. In this study, the authors perform Pb isotope analysis in the EDC ice core, in order to evaluate the dust provenance changes and the associated changes in atmospheric conditions between different climatic conditions before and after the MBE. The manuscript is well written and can represent an important contribution to the scientific literature. However, I believe this manuscript should be published after a minor revision.
Minor Revision:
The results of the mixing model that the authors use, perform large standard deviations compared with the mean values. Even though these large deviations are statistical acceptable, lead to a bad data description. Therefore, I would suggest presenting the median values of their samples and the range for better description of their results.
Response: We agree with the reviewer’s advice. According to this comment, we have replaced the mean values with the median values in Table S3 and S4. Figure 3 was also modified with the median values. In fact, no large differences between mean and median values are observed, except for the contribution from PAP dust during the post-MBE cold periods, for which the mean and median contributions amounted to be 15% and 4.4%, respectively. According to this situation, we have removed the sentences related to this situation (lines 386 and 391-403 in the revised manuscript).
This manuscript is a resubmission of an earlier submission. The following is a list of the peer review reports and author responses from that submission.